# Alcohol use among Congolese Babembe male refugees in Tarrant County: A qualitative study

Elvis Longanga Diese[1,2]◉*, Amy Raines-Milenkov[1,2]◉, Martha Felini[1,2]◉, Idara Akpan[1,2]◉, Arbaz Hussain[1,2]◉, Eva Baker◉[1,2]◉

1 Department of Pediatrics and Women's Health, University of North Texas Health Science Center, Fort Worth, TX, United States of America, 2 Texas College of Osteopathic Medicine, University of North Texas Health Science Center, Fort Worth, TX, United States of America

◉ These authors contributed equally to this work.
* Elvis.LongangaDiese@uth.tmc.edu

**Data Availability Statement:** Data cannot be shared publicly because of limitations of the participant consents approved by the North Texas Regional Institutional Review Board. The de-

## Abstract

### Background

Refugees are at high risk of alcohol abuse due to experiences in their country of origin, transit camps, and in host countries. Congolese have been the largest group of refugees resettled in the US since 2016 and Babembe represent one of the largest Congolese refugee sub-groups. There is a growing body of literature highlighting substance abuse among refugees resettled in the US, but little is known about Congolese Babembe. This study aimed to explore factors and practices contributing to alcohol abuse among Congolese Babembe refugees.

### Methods

A qualitative research design employing a narrative inquiry approach was used in this study. One focus group discussion was conducted with a semi-structured guide in February 2020. A total of 19 hard-to-reach male refugee participants were recruited through snowball sampling. Audio recordings were translated and transcribed before a thematic content analysis was conducted using Nvivo 10.

### Results

The main themes arose in the focus group analysis. First, the role of war, trauma, and migration in the onset of alcohol abuse in a community that once limited alcohol use to only older men for socialization. Second, refugees viewed alcohol abuse in their community as a result of post- resettlement cultural loss. Third, refugees perceived alcohol as serving a purpose to help cope with stressful conditions in the US and bad news received from loved ones in Africa. Further analysis shows the role of interactions with armed forces and other cultures during war and migration contributed to the adoption of alcohol abuse behavior.

identified minimal data set will be available if requested from Dr. Brian A Gladue, Vice President for Research, Institutional Official, Director of the North Texas Regional IRB brian.gladue@unthsc. edu.

**Funding:** This study was funded by Cancer Prevention and Research Institute of Texas (PP170012). The funders had no role in study design, data collection, and analysis, decision to publish, or preparation of the manuscript.

**Competing interests:** No, The authors have declared that no competing interests exist.

## Conclusion

Findings from this study suggest that factors contributing to alcohol abuse among Congolese Babembe refugees include personal traumatic experiences, loss of cultural identity, and conducive conditions in the host country. Understanding these factors can guide the development of appropriate interventions to prevent alcohol abuse in this vulnerable community. Further research is needed to include Babembe women's perspectives.

## Background

The World Health Organization (WHO)'s Global Status Report on Alcohol and Health 2018 states that in 2016 the harmful use of alcohol resulted in 3 million deaths worldwide and 132.6 million disability-adjusted life years (DALYs). According to the report, alcohol is a significant contributor to poor health outcomes in maternal and child health, infectious diseases, injuries, and non-communicable diseases [1]. Drinking alcohol increases the risk of cancers of the mouth, esophagus, pharynx, and larynx, liver, and breast [2].

Refugees or forcibly displaced persons may have risk factors such as grieving the loss of their home countries and escaping violence, both associated with a higher risk of alcohol abuse [3]. Several studies have indicated that people who have had traumatic life events have an increased risk of alcohol abuse [4–6]. For example, one study found that refugees may drink to cope with trauma, boredom, and frustration, as well as a social experience with others [7]

The Babembe or Bembe are an ethnolinguistic group from the province of South Kivu in the eastern Democratic Republic of Congo [8]. No official statistics are available on the size of the Babembe population. One source estimates around 600,000 Babembe living in eastern Congo while another 30,000 in Tanzania [9]. The South Kivu province, which borders Rwanda, Burundi, and Tanzania, has been particularly affected by various armed conflicts in the Congo that have led to or contributed to an estimated 6 million deaths [10, 11]. Multiple armed conflicts in the Congo have been ongoing since the 1994 Genocide that reignited ethnic rivalries in the neighboring Congo. Armed Babembe militia groups have since then been fighting against the Banyamulenge militiamen for the control of lands in the Fizi territory of South Kivu [8, 12]. Two decades of armed conflicts in Congo have resulted in a massive displacement of millions of Congolese refugees to neighboring countries or as internally displaced persons. Babembe families who have fled violence in Congo have found temporary protection at the United Nations' refugee camps mainly in Kigoma, Tanzania [8]. From refugee camps, Babembe families have been resettled in the United States in areas such as North Texas and Boise Idaho [13]. In recent years, thousands of refugees from the Congo have been resettled in the US. In fact, since the fiscal year 2016, Congolese refugees have represented the largest number of refugee admissions in the country [14]

Few studies address how to reduce the risk of alcohol abuse in refugee populations [6, 7]. Among the limited studies addressing alcohol use and abuse in refugee communities, none were found to focus on the Babembe despite this group being a high-risk community for alcohol abuse [15]. The Building Bridges Initiative (BBI), a community health worker cancer prevention services intervention and research program serving the North Texas refugee community, has identified high alcohol consumption among both male and female members of the community with consequences to family and health outcomes. The purpose of our study was to explore factors and practices contributing to alcohol abuse among the Congolese Babembe refugee community. Understanding those factors will guide the development of appropriate interventions to prevent alcohol abuse in this vulnerable community.

## Methods

This exploratory study was designed to investigate Babembe refugee resettlement experiences and their beliefs, experiences, and practices of alcohol use. Data was collected using a single focus group discussion using a semi-structured guide with 19 participants living in Tarrant County, Texas. Subjects in this study were recruited through snowball recruitment. Most of the focus group participants were referred by Babembe's community leader who lives in the same apartment complex housing the majority of Babembe refugees. Entry into the community was achieved through gaining the approval of the study by this community leader. The community leader shared information to an estimated 45 community members about the purpose, date, and location of focus groups using WhatsApp group, a free cell phone cross-platform messaging and voice service that the Babembe community in Tarrant County already uses to regularly disseminate information. Of those messaged in the community group, 18 agreed to participate in the focus group. One of the 19 focus group participants was a current Building Bridges participant who had agreed to be re-contacted for additional research studies. Service records of Building Bridges participants were reviewed to identify eligible participants who had consented to be recontacted for further studies. Eligible participants were contacted to determine their willingness to participate in a small group discussion. Those that did not want to participate primarily cited work schedule conflict as the reason to not attend the discussion.

Trained bilingual research personnel conducted the consent process, study procedures, surveys, and focus groups with the assistance of certified interpreters. Key personnel fluent in the common languages of this community, French, Lingala, and Swahili conducted the consenting process. Due to the limited literacy among Congolese refugees from South Kivu [14], the consent form was read and explained in simple language to all participants before seeking verbal consent from them. Participants were given an opportunity to ask questions about the study during the verbal consenting process.

The focus group discussion took place in February 2020. Data collection occurred in two parts on the same day of focus group discussion. Prior to the focus group discussion, key personnel collected baseline sociodemographic and behavioral information from consented participants using a demographic paper survey. The paper survey included 12 questions on participant's age, ethnicity, and education, marital status and family information, insurance status, length of time in the United States, information about children, and alcohol use.

We conducted a focus group discussion with 19 adult men from the Babembe refugee community living in Tarrant County, Texas. Though there may be some saturation in analysis, this community felt more comfortable speaking up in larger numbers for increased protection from identification. Table 1 describes the sociodemographic profile of participants. The mean age of participants was 47 years (range: 25 years—68 years); 68% never completed high school. Participants spent on average 21 years in refugee camps before being resettled in the US. Only consented participants attended the focus group discussion.

Fifteen prompts using open-ended questions included topics of arrival in the US, needs, alcohol use prior to arrival in the US, alcohol use in the US, and concerns with alcohol use. Each participant in the focus group received a $20.00 gift card to a local grocery store for their time. The focus group was conducted in the respective language of the participants and recorded with an audio recorder. Recordings were transcribed and translated by trained and authorized key research personnel, a professional interpreter. All identifying information was removed from the transcripts.

Using Nvivo 10 and thematic content analysis, a team reviewed common themes and discussion to narrow down topics to three main themes and eleven sub-themes. Reviewers

**Table 1. Sociodemographic profile of participants.**

| Characteristic | Total n (%) |
|---|---|
| **Gender** | |
| Male | 19 (100) |
| **Age (in years)** | |
| 29 or below | 2 (10) |
| 30–39 | 5 (26) |
| 40–49 | 2 (11) |
| 50–59 | 6 (32) |
| 60 or above | 4 (21) |
| **Marital Status** | |
| Single | 1 (5) |
| Married | 17 (89) |
| Widowed | 1 (5) |
| **Education level** | |
| Less than high school diploma | 13 (68) |
| High school Diploma | 4 (21) |
| More than high school | 2 (11) |
| **Years lived in a refugee camp** | |
| 15–19 years | 2 (11) |
| 20 years and more | 17 (89) |
| **Years lived in US** | 19 (100) |
| Less than 5 years | 17 (89) |
| 5 years and more | 2 (11) |
| **Number of the children in the household** | |
| None | 2 (11) |
| 1–2 | 1 (5) |
| 3–4 | 4 (21) |
| 5 or more | 12 (63) |

included key personnel who helped conduct the focus group and the three other staff members in the BBI program for whom the analysis was blinded.

To address validity, findings were reviewed with the community members at the end of the focus group discussion. Key points from the discussion were shared with participants for confirmation. Key personnel received feedback on the statements from participants. This is a validity technique in qualitative research known as a Member Check [16].

Authors of the study have several years of combined experience in analyzing qualitative data, working with refugee communities in the US, as well as working with vulnerable populations in eastern Congo.

This study was approved by the North Texas Regional Institutional Review Board.

## Results

Three themes and nine subthemes emerged from the data.

### Theme 1: Role of war, trauma, and migration

**Subtheme 1: War and forced migration introduced alcohol abuse.** The majority of participants viewed alcohol abuse as a foreign practice introduced among the Babembe by

government soldiers and other tribes after the onset of war in the mid-1990s in Easter Congo as illustrated by the following quotes:

"*It was mostly because of the influence of the army. During their operations in our territories, the soldiers initiated people to alcoholism and since then, it has never stopped. Alcohol abuse was not part of the Babembe culture, it is a new phenomenon in our society which was brought about by the military presence in our area*".

"*We blame the alcohol abuse on the conflicts in the Great Lake Region comprising the eastern DR Congo, Rwanda, and Burundi. As people were fleeing the conflicts from one country to the other, People were mixing and copying from each other's culture. And as mentioned by the previous speaker, the army is made of several tribes, many of whom are alcohol prone and they also brought that habit in the Fizi territory*"

Most participants also described that prior to the war, alcohol abuse was not tolerated in the Babembe culture and only adult men were allowed to moderately drinking alcohol as shared by the following participant:

"*In the Bembe culture, one was allowed to take alcohol only after marriage. I saw a relative of mine being beaten in public because of alcohol abuse. He got drunk and started insulting people. The community beat him seriously to correct him.*"

**Subtheme 2: Alcohol abuse addressed in refugee camps.** Several participants explained that alcohol abuse was already a problem in refugee camps prior to resettlement in the US. One participant described his participation in one initiative to address alcohol abuse in a refugee camp:

"*In Tanzania, I was the leader of a youth group called TKK. I led this group for 8 years. In this group, we had a session called the consumption of intoxicating products [Alcohol and drugs]. When he [the instructor] came to teach us. . . he brought different types of alcoholic beverages So, we went on drinking. . . He said they got an alarming report that the youth in the camp are abusing alcohol, cigarettes, drugs, and other intoxicating products. He went on with his lesson up to the end then he asked us what is pushing us to abuse the consumption of alcohol? He got good answers because everyone had a bottle of alcoholic beverage*"

**Subtheme 3: Past trauma and family separation.** Past trauma and stress from family separation were identified by most participants as contributing factors to alcohol consumption and abuse after resettlement in the US.

For example, one participant explained: "*What I want to add here is this: I know we are trying to hide our frustration, the trauma of losing the entire family during the war back home for example.*"

Citing similar trauma, another participant added "*I have a sister who was beaten in my country until she got crippled. We went through the resettlement process together hoping that we were going to travel together, alas, she remained in the refugees' camp and now she got impregnated twice and my mother who was looking after her passed on.*"

**Subtheme 4: Post-resettlement ease of access to alcohol.** Easy access and affordable alcohol beverages in the US were also viewed as contributing to alcohol consumption among refugees.

"...here there are many types of alcoholic beverages with different levels of alcoholic content. What makes the difference is that there it was very difficult to get money but here people easily get the money and the beverages are quite cheap. With one Dollar you can get your beverage. Sometimes people don't care about coins, so the alcohol consumer can easily gather those coins and buy himself a drink."

Another participant similarly described the situation in the US as:

"*Alcohol is consumed like water here in the US. Wherever you go, you will find alcohol, maybe it's because it is so cheap.*"

## Theme 2: Post-resettlement cultural loss

**Subtheme 5: America has destroyed our traditional ways to control alcohol abuse.** Participants explained that alcohol abuse in their community was associated with the loss of their cultural identity resulting from their migration to the US.

"*Traditionally, in Babembe culture. . . the child becomes an adult at the age of 20. If at 18 years of age a child is caught drinking alcohol, they will be seriously beaten by the community. But today we see our children taking alcohol at the age of 18 years because they were told that at that age they are mature to do whatever they want. You have destroyed our culture. If we had brought up our kids according to our culture, we could be having well-behaved children.*"

Participants also blamed the rise in alcohol abuse among teenagers on the parents' loss of the ability to punish their children out of fear of law enforcement. Participants blamed the government for taking away their parental ability to punish their children who are abusing alcohol.

"*We know where we come from, when we were growing up, it was the elders who were drinking alcoholic beverages. And if as a young man you taste alcohol you will be afraid because you know that you could be severely punished. But here in the USA, the rules have destroyed our children and wives. Because they know that they cannot be punished. They are free, they have the money and parents have nothing to tell them or else 911.*"

Another participant similarly expressed:

"*At 18 you cannot tell him/her anything. If you try to interfere with his/her life, 911*"

One participant expressed his frustration by regretting bringing his family to the US: "*At 18, a child should still be under parents' control. We regret bringing our children to this country.*"
An unmarried young adult participant similarly endorsed this theme: "*We are abusing the consumption of alcohol because we don't fear our parents anymore. We know that they won't touch us. Because we are in America.*"

**Subtheme 6: American culture and policies contribute to alcohol abuse.** Most participants also blamed the American culture and policies for limiting parents and community's influence over their youth on American laws they believe are limiting the community's ability to use traditional methods to discipline members who abuse alcohol.

"*Here in the USA, the responsibility of child behavior lies in the hands of the government. Bad relationships between parents and children are because of government policies. The freedom*"

*rule of setting a child free from parents at the age of 18 is wrong and secondly, a child who grew up in the USA up to the end of secondary school is no longer your child, you can't tell him anything. If you dare, you are shown the telephone. And the government punishes the parent, not the child. That is a shame in our culture.*"

Similarly, another participant stated:

"*We blame the government policies that encourage children not to listen to parents' rules. At 18, a child should still be under parents' control. We regret bringing our children to this country.*"

## Theme 3: Alcohol serves a purpose

**Subtheme 7: Alcohol is food and medicine and socialization.**    Most participants agreed that alcohol had a functional use. They viewed alcohol as good for health, describing it either as food or medicine, as explained by the following participants: "*Firstly, alcohol is food. You hear people share beverages saying "To your health" because alcohol is food and it has some good effects on the body.*"
Similarly:

"*Alcohol plays a medicinal role in our body as said by medical doctors. There are people who were told by their Doctors that taking 1 or 2 glasses of wine per day is very acceptable.*"

Another participant stated:

"*We agreed that consumed with moderation, alcohol is food. In our village, we mix modern beverages with our traditionally brewed beverages (Mmena) to soften the alcohol content. Even some medicinal plants are mixed with the same concoction to produce better effects.*"

Most participants described alcohol consumption for socialization purposes. Describing the appropriate time to consume alcohol, one participant said: "It's better around 6 pm when you have a few bottles to go and see a friend, sit and share, talk and discuss."
Another participant described the traditional context for alcohol consumption in the Babembe culture in the following way: "In the area where we originate, alcohol used to be consumed by adults and was consumed with moderation and for socialization amongst relatives or friends."
**Subtheme 8: Belief that alcohol helps to deal with work-related stress and bad news from Africa.**    Hard labor in the US and bad news received from relatives in Africa are stressors that lead refugees to consume alcohol as illustrated by the following quotes:

"*You know how people are working nowadays. You spend the day at work, standing for 10 to 12 hours, by the time you return home, you are in pain, exhausted. As you sit down on your sofa, you don't want to move. If you call most people after working hours, they will tell you they aren't in bed but on the sofa, exhausted. Because of that, they will look for something to energize them; one or two cold ones (alcohol) from the refrigerator to cool down*"

Another participant explained:

"*Sometimes you will have spent a very bad day with your supervisor when you reach home, you want to forget that situation. You could have quarreled with a colleague at work or a*

*friend or even some bad news from relatives in Africa: 'send us some money, we need this and that, so and so are sick and need to be hospitalized,' all these contribute to one wanting to take some alcohol and reduce the stress.*"

**Subtheme 9: Alcohol increases lack of control.**    Alcohol was described as making individuals feel emboldened or more likely to act in a way that they would have not otherwise.

"*If you drink and stop at level 1, you cannot beat your wife. If you drink and stop at level 2, you cannot beat your wife. But when you consume alcohol and reach level 3 of drunkenness, that is the level where you feel as being the strongest man on earth, the richest man and when you go home, any little incident will lead you to fight and beat your wife. That is the level where you are controlled by alcohol instead of you controlling it. If alcohol says beat this one, you just do that.*"

## Discussion

This study explored factors and practices contributing to alcohol use and abuse among the Congolese Babembe refugee community. Three major themes emerged to describe how war/forced migration, loss of cultural identity, and the belief in functional use of alcohol have contributed to alcohol abuse among Congolese Babembe refugees.

Although several themes emerged, the underlying and unifying theme was the role of conflict-induced displacement and the resulting cultural identity loss on alcohol abuse among Congolese Babembe. Our findings suggest that while alcohol has been traditionally consumed by the Congolese Babembe for different functional roles, the onset of alcohol abuse was associated with war and forced migration from armed conflicts. The harmful consumption of alcohol by the Babembe refugees was later amplified by a sense of loss of cultural identity and environmental factors after their resettlement in the US.

Findings from this study linking forced migration and alcohol abuse are consistent with existing literature. Refugees often experience conflict-related trauma that increases their risks for mental health disorders and alcohol abuse [17–21]. Prolonged stay in harsh refugee camp settings has also been associated with substance abuse among refugees [19]. According to the UNHCR, only 1% of refugees are accepted into resettlement countries, and on average they spend about 17 years in a refugee camp before being resettled in a new country [22]. Most Congolese Babembe refugees enrolled in the BBI program spent at least 21 years in refugee camps in Tanzania and Namibia before their resettlement in the US.

Loss of cultural identity is a stress-inducing factor that has been associated with alcohol abuse among refugees. Refugees resettled in Western countries have greater risks to engage in alcohol abuse as they go through the acculturation process [19]. Stress associated with new challenges such as learning a new language, adjusting to a new culture, and finding a job increased the risks of alcohol abuse. In addition, the acculturation process is often associated with a disruption of traditional cultural structures resulting from a change in the balance of power as women and young people become financially independent [19]. Young refugees are particularly vulnerable to loss of cultural identity. Coming from conservative societies where they are not allowed to consume alcohol, young refugees experience the pressure to conform to their new environment where alcohol consumption is normalized [17].

Environmental factors in host countries also play an important role in alcohol abuse among refugees. Similar to findings from our study, refugees resettled in Western countries perceive their new environments to be conducive to alcohol abuse. In most cases, alcohol is widely available and easily accessible to newly resettled refugees [17]; offering them a readily available coping mechanism to deal with post-traumatic stress and acculturation challenges. Babembe

refugees in North Texas have mainly been resettled in neighborhoods where alcohol and drug abuse is prevalent [23].

## Conclusion

The findings of this study show that alcohol abuse among the Congolese Babembe refugees is a result of both pre and post resettlement factors. Personal traumatic experiences, loss of cultural identity and the ease of access to alcohol are the main contributing factors identified in this study. Alcohol and substance abuse is a growing concern among refugees resettled in the US and these findings may contribute to improved resources and support services for incoming refugees. Past trauma and the loss of cultural identity highlight the need to have long term resources to support mental health support groups for refugees. We also recommend newly arrived refugees begin to be offered housing located in neighborhoods that are less enabling of substance abuse.

### Methodological considerations

This focus group was larger than traditional 8–10 participants, which could lead to saturation. However, this community is highly collective and seemed to build strength in speaking up by the larger number of participants. It would be good to test focus group sizes in similar groups with collectivistic communities.

Additionally, with cultural differences, there can be a social desirability bias issue. It was interesting to find that in the demographic form that was completed by each participant, all noted that they did not drink any alcohol. However, in the discussion participants were open about their alcohol use. We addressed the social desirability bias in our discussion through the key personnel who were not from the same subgroup but originated from the same country of origin. We also scheduled a trusted interpreter from the same community and included the tribal leader in the entire process for approval of the discussion. We limited our use of paper forms because of distrust in the system and explained before the discussion that it would all be deidentified.

Further research should include women from the same community on alcohol use. Using trusted female interpreters, and key personnel from similar backgrounds to facilitate a female focus group on alcohol to replicate the male focus group discussion. A broader view of the social determinants and issues around racism would also be helpful to review for impact on substance use in this group.

## Author Contributions

**Conceptualization:** Elvis Longanga Diese, Amy Raines-Milenkov, Eva Baker.

**Data curation:** Elvis Longanga Diese.

**Formal analysis:** Amy Raines-Milenkov, Martha Felini, Idara Akpan.

**Funding acquisition:** Amy Raines-Milenkov, Eva Baker.

**Methodology:** Martha Felini.

**Project administration:** Eva Baker.

**Software:** Elvis Longanga Diese.

**Supervision:** Amy Raines-Milenkov, Eva Baker.

**Validation:** Elvis Longanga Diese, Martha Felini.

**Writing – original draft:** Elvis Longanga Diese, Eva Baker.

**Writing – review & editing:** Amy Raines-Milenkov, Martha Felini, Idara Akpan, Arbaz Hussain.

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
