## [Decision Letter · Decision Letter 0]

10 Jan 2022

PONE-D-21-33135Alcohol use among Congolese Babembe refugees in Tarrant County: A qualitative studyPLOS ONE

Dear Dr. Baker,

Thank you for submitting your manuscript to PLOS ONE. After careful consideration, we feel that it has merit but does not fully meet PLOS ONE’s publication criteria as it currently stands. Therefore, we invite you to submit a revised version of the manuscript that addresses the points raised during the review process.

I may inform you that the reviewers have several important points that needs the authors' systematic response. However, your submission of the manuscript  after due revision does not guarantee an acceptance of the manuscript for publication. 

We look forward to receiving your revised manuscript.

Kind regards,

Subhendu Kumar Acharya, Ph.D

Academic Editor

PLOS ONE

Journal Requirements:

“No. The funders had no role in study design, data collection and analysis, decision to publish, or preparation of the manuscript”

Reviewers' comments:

Reviewer's Responses to Questions

**Comments to the Author**

1. Is the manuscript technically sound, and do the data support the conclusions?

Reviewer #1: Yes

Reviewer #2: Partly

2. Has the statistical analysis been performed appropriately and rigorously? 

Reviewer #1: N/A

Reviewer #2: N/A

3. Have the authors made all data underlying the findings in their manuscript fully available?

Reviewer #1: Yes

Reviewer #2: Yes

4. Is the manuscript presented in an intelligible fashion and written in standard English?

Reviewer #1: Yes

Reviewer #2: Yes

5. Review Comments to the Author

Reviewer #1: The purpose of this study is to explore the determinants and practises that contribute to alcohol use among one of the world's most vulnerable populations. While this work has major implications for public health, it should clarify or address the following points.

The title of the article is 'Alcohol usage among Congolese Babembe refugees in Tarrant County: A qualitative study.' Kindly include among male refugees, as this study is mostly focused on male participation.

Abstract: Indicate in approach which approach you used, such as phenomenology, ethnography, or narrative. Similarly, make reference to the analysis method of 'thematic content analysis.'

Introduction: There are no comments.

Methods:

It stated unequivocally that 19 individuals were chosen using snow-ball sampling. How many focus groups were held by the authors? Each FGD recorded the number of participants. I believe that facilitating 19 participants in a single FGD is tough. How the author dealt with it. All of these approaches put limits on your data collection procedures. Alternatively, the author may refer to 19 interviews conducted with semi-structured open-ended questions.

How many individuals did you approach? And how many are unwilling to participate, and what are their reasons? Are there any additional non-participants present during the FGDs?

Kindly include one sentence describing each author's educational background and professional experience in qualitative research, as well as how their multidisciplinary backgrounds help to extend the scope of the findings' interpretation. You may mention investigator triangulation here.

Results:

The participants characteristics you can move to methods section. Since it is only 19 participants instead of table you may summarize the key participants characteristics.

In each quotation mention the age and years lived in refugees camp of the participant.

In Subtheme 2: Alcohol abuse in refugee camps. Reduce the quotation length and describe the content in detail.

The ‘Subtheme 5: America has destroyed our culture, and Subtheme 6: Loss of traditional ways to control alcohol abuse’ may be combined.

Similarly, sub-theme 8 to 11 may be classified as two broad sub-themes.

There is need of table on coding tree. Example of coding tree given below.

Themes

Sub-themes

Codes

Discussion:

Mention one paragraph on ‘Implication for policy and practice’

Instead of limitations, put the heading as ‘Methodological considerations’ and discus both strength and limitations of the study.

Conclusion section is missing.

Reviewer #2: The central contribution of the manuscript is to help the reader understand the factors contributing to alcohol abuse among the Babembe refugees who are originally from Congo. While the empirical material and analysis indicate the relationship between the refugee status and alcohol abuse, it may require more nuanced analysis of the data to present the argument better. The connection between the theoretical argument and the empirical material needs to be strengthened

Some specific comments:

1. The authors should include a justification of why FGD was conducted with 19 members in the methods section. While they discuss this in the limitations, it may help the reader to contextualise the results if it is mentioned upfront.

2. FGD using a semi-structured guide make better sense than mentioning ‘semi-structured FGD’.

3. The analysis section should be strengthened by including a description of the analysis process and epistemological stance of the researchers in choosing thematic content analysis. They should also provide a reference for the analysis framework.

4. Use of identifiers that do not reveal the identities of the respondents along with the quotations is important.

5. For the subtheme 2, the quote on TKK does not seem to fit into the theme of alcohol abuse. The quote could be used to substantiate the existing efforts to tackle the alcohol abuse problem and the community participation.

6. Subtheme 4 indicates ease of access and low cost of alcohol. The theme can be rephrased to effectively communicate to the reader

7. Under theme 2, all the subthemes are overlapping. There has to be more nuanced presentation of the data in this theme.

8. Are the themes and subthemes based on the verbatim from the FGD?

9. Theme 3 and the sub themes need to be rephrased.

10. Sub-theme 10 is misleading about empowerment. The statement and the quote convey that beating wife is empowerment for men. I’m sure that that the community and the researchers do not subscribe to this!!

11. The discussion could be have been more effective if discussion around larger social determinants, issues around racism shape the experience of the refuges in US.

12. The manuscript should have included some discussion around how these factors were dealt with among other refugees or could be dealt with through appropriate interventions among the Babembe.

6. PLOS authors have the option to publish the peer review history of their article (what does this mean?). If published, this will include your full peer review and any attached files.

Reviewer #1: **Yes: **Krushna Chandra Sahoo

Reviewer #2: **Yes: **Nanda Kishore Kannuri

---

## [Author Response · Author response to Decision Letter 0]

8 Mar 2022

We have updated all the changes that were listed in the review and have attached a letter to the reviewers to show details of the adjustments. Thank you so much for your work.

---

## [Decision Letter · Decision Letter 1]

18 Jul 2022

Alcohol use among Congolese Babembe male refugees in Tarrant County: A qualitative study

PONE-D-21-33135R1

Dear Dr. Baker,

We’re pleased to inform you that your manuscript has been judged scientifically suitable for publication and will be formally accepted for publication once it meets all outstanding technical requirements. We apologize for the delay incurred on your submission. 

Kind regards,

Dario Ummarino, PhD

Senior Editor

PLOS ONE

Additional Editor Comments (optional):

Reviewers' comments:

Reviewer's Responses to Questions

**Comments to the Author**

1. If the authors have adequately addressed your comments raised in a previous round of review and you feel that this manuscript is now acceptable for publication, you may indicate that here to bypass the “Comments to the Author” section, enter your conflict of interest statement in the “Confidential to Editor” section, and submit your "Accept" recommendation.

Reviewer #1: All comments have been addressed

2. Is the manuscript technically sound, and do the data support the conclusions?

Reviewer #1: Yes

3. Has the statistical analysis been performed appropriately and rigorously? 

Reviewer #1: N/A

4. Have the authors made all data underlying the findings in their manuscript fully available?

Reviewer #1: Yes

5. Is the manuscript presented in an intelligible fashion and written in standard English?

Reviewer #1: Yes

6. Review Comments to the Author

Reviewer #1: In the manuscript 'Alcohol use among Congolese Babembe male refugees in Tarrant County: A

qualitative study' thanks for addressing all the comments.

7. PLOS authors have the option to publish the peer review history of their article (what does this mean?). If published, this will include your full peer review and any attached files.

Reviewer #1: **Yes: **Dr. Krushna Chandra Sahoo

---

## [Editor Report · Acceptance letter]

25 Jul 2022

PONE-D-21-33135R1 

Alcohol use among Congolese Babembe male refugees in Tarrant County: A qualitative study 

Dear Dr. Baker:

I'm pleased to inform you that your manuscript has been deemed suitable for publication in PLOS ONE. Congratulations! Your manuscript is now with our production department. 

Kind regards, 

on behalf of

Dr Dario Ummarino, PhD 

Staff Editor

PLOS ONE